# Ethnic disparities in COVID-19 mortality in Mexico: A cross-sectional study based on national data

**Ismael Ibarra-Nava** , **Kathia G. Flores-Rodriguez** , **Violeta Ruiz-Herrera** , **Hilda C. Ochoa-Bayona** , **Alfonso Salinas-Zertuche, Magaly Padilla-Orozco** , **Raul G. Salazar-Montalvo**  *

Department of Preventive Medicine and Public Health, Faculty of Medicine, Universidad Autónoma de Nuevo León, Monterrey, Nuevo Leon, Mexico

* raul.salazarm@uanl.mx

## Abstract

**Data Availability Statement:** We have attached a zip file containing the csv file (data set) with the raw data. This data set was downloaded directly

### Introduction

Across the world, the COVID-19 pandemic has disproportionately affected racial and ethnic minorities. How ethnicity affects Indigenous peoples in Mexico is unclear. The aim of this cross-sectional study was to determine the mortality associated with ethnicity, particularly of Indigenous peoples, in a large sample of patients with COVID-19 in Mexico.

### Methods

We used open access data from the Mexican Ministry of Health, which includes data of all confirmed COVID-19 cases in the country. We used descriptive statistics to compare differences among different groups of patients. Logistic regression was used to calculate odds ratios while adjusting for confounders.

### Results

From February 28 to August 3, 2020, a total of 416546 adult patients were diagnosed with COVID-19. Among these, 4178 were Indigenous peoples. Among all patients with COVID-19, whether hospitalized or not, a higher proportion of Indigenous peoples died compared to non-Indigenous people (16.5% vs 11.1%, respectively). Among hospitalized patients, a higher proportion of Indigenous peoples died (37.1%) compared to non-Indigenous peoples (36.3%). Deaths outside the hospital were also higher among Indigenous peoples (3.7% vs 1.7%). A higher proportion of Indigenous peoples died in both the private and public health care sectors. The adjusted odds ratio for COVID-19 mortality among Indigenous peoples with COVID-19 was 1.13 (95% confidence interval 1.03 to 1.24). The adjusted odds ratio for COVID-19 mortality among Indigenous peoples with COVID-19 was higher among those who received only ambulatory care (1.55, 95% confidence interval 1.24 to 1.92).

from a Government of Mexico's website and it is open access data. The uploaded file contains data up to August 3rd, 2020. You can download the data at https://datos.gob.mx/busca/dataset/informacion-referente-a-casos-covid-19-en-mexico. However, you will find the data set is updated daily, so it will not be the same as the one we have attached a Supporting Information file.

**Funding:** The author(s) received no specific funding for this work.

**Competing interests:** The authors have declared that no competing interests exist.

## Discussion

In this large sample of patients with COVID-19, the findings suggest that Indigenous peoples in Mexico have a higher risk of death from COVID-19, especially outside the hospital. These findings suggest Indigenous peoples lack access to care more so than non-Indigenous people during the COVID-19 pandemic in Mexico.

## Introduction

As COVID-19 cases and deaths continue to increase around the world, an increasing amount of evidence shows that the current pandemic is disproportionately affecting socially disadvantaged groups. Several studies have demonstrated that older individuals and people with underlying medical conditions have an increased risk for adverse outcomes, including death [1, 2]. Moreover, emerging evidence from the USA, the UK, and Brazil has shown that racial and ethnic minorities (REM) have a higher risk of adverse outcomes and death from COVID-19 as well [3–5]. These health disparities may be due to a higher prevalence of underlying comorbidities and poor access to high quality health care services among REM. Furthermore, specific public health preventive measures to address COVID-19 are hard to implement in REM communities that live in densely populated areas, work on essential services, and use public transportation [6].

Mexico has some of the highest numbers of confirmed cases and deaths in the world. As of August 5th, Mexico ranks seventh worldwide in total number of cases and third in total number of deaths [7]. Access to testing has been a challenge in Mexico, which has one of the lowest testing rates worldwide [8]. Therefore, the true number of cases and deaths is probably higher. Similar to studies conducted in other settings, Mexicans who are older and who have an underlying medical comorbidity have a higher risk of adverse outcomes and death [9–11]. Mexico is a racially and ethnically diverse country, with an indigenous population of over 12 million (10.1% of the total population) [12]. For centuries, indigenous peoples in Mexico have been marginalized and unfairly treated, resulting in unjust health inequities and unequitable access to health care.

The Epidemiological Surveillance System of Viral Respiratory Diseases of the Mexican Ministry of Health (MoH) collects sociodemographic and clinical data of diagnosed COVID-19 cases, across both public and private hospitals. These data are updated daily and are openly available to anyone. In this study, we aimed to analyze the relationship between risk factors, mainly ethnicity, and COVID-19 adverse outcomes (hospitalization and ICU admission) and mortality. The impact of COVID-19 on indigenous health needs be analyzed to address the existing health inequities that persist in Mexico. Furthermore, our results could help local, state, and federal governments identify high-risk groups for severe illness and death and help them with the implementation of specific preventive measures in these socioeconomically disadvantaged groups.

## Methods

### Study design and population

The present cross-sectional study is based on national and open access data reported by the MoH. These data set is published through the Epidemiological Surveillance System for Viral Respiratory Diseases (SISVER) and it includes data on all suspected COVID-19 patients who

were tested for SARS-CoV-2. For this study, we included data from laboratory-confirmed patients with COVID-19 in Mexico from February 28 to August 3rd, 2020. The SISVER reports laboratory-confirmed cases through 475 Viral Respiratory Diseases Monitoring Units (*USMERs*) across the country, which consist of both public and private providers in primary care and hospitals. Patient information is collected by physicians based on an epidemiological study format for all suspected cases of COVID-19. Laboratory-confirmed cases were defined as a positive result of real-time reverse transcriptase-polymerase chain reaction (RT-PCR) assay.

A total of 443813 laboratory-confirmed COVID-19 cases were included in the SISVER data set. We excluded 11312 patients who were 17 years or younger, 12562 patients whose ethnicity was not recorded and another 3393 patients with missing or unknown data for type of health care sector (public or private) in which they were treated for COVID-19. The total sample for our study consisted of 416546 patients. We further divided our sample by hospital admission, resulting in two samples of 113853 hospitalized patients and 302693 non-hospitalized patients. Missing or unknown data on comorbidities and risk factors were treated as negative for that comorbidity or risk factor as they were most likely left blank on the form if the patient did not have it or did not know they had it. The variables included in our study were sociodemographic characteristics (age, gender, ethnicity, and health care sector), comorbidities (diabetes, chronic obstructive pulmonary disease, high blood pressure, and chronic kidney disease), risk factors (obesity and smoking status), and adverse outcomes (hospital admission, ICU admission, endotracheal intubation, and death).

In Mexico, three different definitions are used to determine if a person is Indigenous [13]. The most common one is the population aged 3 or older that speaks an Indigenous language; this represents 6.5% of the population. The second definition is living in a household where at least one person speaks an Indigenous language; this represents 10.1% of the population. Finally, the last definition is about self-identity. Around 21.5% of Mexicans self-identify as Indigenous. In our data set, ethnicity was assessed by asking if the patient spoke any indigenous language, regardless of whether they also spoke Spanish or not. These data are openly available at https://coronavirus.gob.mx/.

## Statistical analysis

Continuous variables were described using the mean and standard deviation. Categorical variables were described in frequencies and percentages. We estimated odds ratios (ORs) with 95% confidence intervals (95%CIs) and their corresponding *p-values* for death by sociodemographic characteristics, comorbidities and risk factors. A multiple logistic regression model was used adjusting by age, gender, and health care sector for each of the comorbidities and risk factors previously mentioned. The selection of variables for our model was based on a review of the literature. All statistical analysis was performed using R version 4.0.2 and RStudio Desktop version 1.3.1073. No ethics approval was required for this study.

## Results

The MoH COVID-19 database included 1011050 suspected cases of COVID-19 as of August 3rd, 2020. Of these, 432501 adult patients tested positive for COVID-19. A total of 12562 had missing data on their ethnicity and 3393 had missing data on the type of health care sector (private or public) where they received treatment. In the end, our sample consisted of 416546 adult patients with COVID-19 (Fig 1). Of these, 113853 required hospitalization (hospitalized sample) and 302693 received only ambulatory care (non-hospitalized sample).

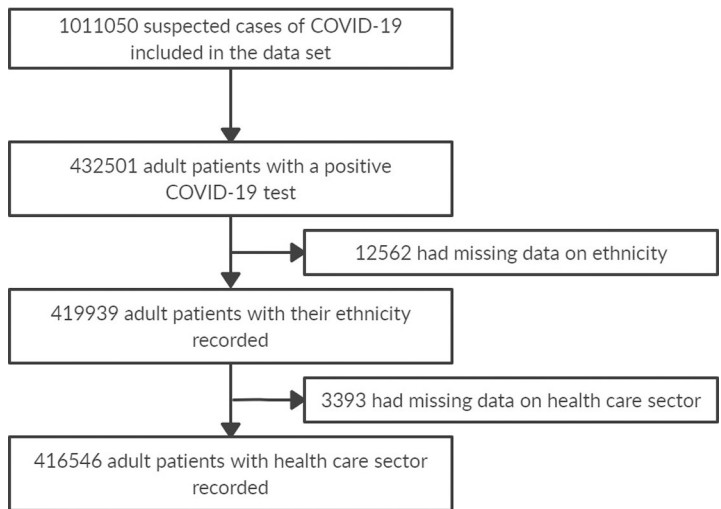

**Fig 1. Flowchart of the MoH data set used for this study.**

Table 1 shows the demographic, comorbidity, and risk factor data among survivors and non-survivors in all three of our samples. Across all samples, the mean age of non-survivors was similar (61.0 in non-hospitalized patients and 62.0 in hospitalized patients) and higher than survivors. Among survivors, the mean age of hospitalized patients was higher than non-hospitalized patients (53.6 and 41.8, respectively). Across all samples, a higher proportion of

**Table 1. Sociodemographic characteristics and present comorbidities and risk factors among survivors and non-survivors of all COVID-19 patients, hospitalized patients and non-hospitalized patients.**

| | All COVID-19 patients (n = 416546) | | Hospitalized patients (n = 113853) | | Non-hospitalized patients (n = 302693) | |
|---|---|---|---|---|---|---|
| | Survivors n = 370038 (88.8%) | Non-survivors n = 46,508 (11.2%) | Survivors n = 72566 (63.7%) | Non-survivors n = 41287 (36.3%) | Survivors 297472 (98.3%) | Non-survivors 5221 (1.7%) |
| Age (years ± SD) | 44.1 ± 14.7 | 61.9 ± 13.8 | 53.6 ± 15.0 | 62.0 ± 13.8 | 41.8 ± 13.6 | 61.0 ± 14.1 |
| Gender | | | | | | |
| Women (n = 195153) | 178795 (91.6%) | 16358 (8.4%) | 29357 (66.8%) | 14588 (33.2%) | 149438 (98.8%) | 1770 (1.2%) |
| Men (n = 221393) | 191243 (86.4%) | 30150 (13.6%) | 43209 (61.8%) | 26699 (38.2%) | 148034 (97.7%) | 3451 (2.3%) |
| Ethnic group | | | | | | |
| Non-indigenous (n = 412368) | 336551 (88.9%) | 45817 (11.1%) | 71559 (63.7%) | 40692 (36.3%) | 294992 (98.3%) | 5125 (1.7%) |
| Indigenous (n = 4178) | 3487 (83.5%) | 691 (16.5%) | 1007 (62.9%) | 595 (37.1%) | 2480 (96.3%) | 96 (3.7%) |
| Sector | | | | | | |
| Private (n = 11476) | 10911 (95.1%) | 565 (4.9%) | 2579 (84.2%) | 485 (15.8%) | 8332 (99.0%) | 80 (1.0%) |
| Public (n = 405070) | 359127 (88.7%) | 45943 (11.3%) | 69987 (63.2%) | 40802 (36.8%) | 289140 (98.3%) | 5141 (1.7%) |
| Comorbidities | | | | | | |
| Diabetes (n = 68137) | 50477 (74.1%) | 17660 (25.9%) | 20773 (56.8%) | 15794 (43.2%) | 29704 (94.1%) | 1866 (5.9%) |
| COPD (n = 6633) | 4394 (66.2%) | 2239 (33.8%) | 2067 (51.2%) | 1972 (48.8%) | 2327 (89.7%) | 267 (10.3%) |
| High blood pressure (n = 84577) | 64194 (75.9%) | 20383 (24.1%) | 22992 (55.8%) | 18184 (44.2%) | 41202 (94.9%) | 2199 (5.1%) |
| Chronic kidney disease (n = 8444) | 5264 (62.3%) | 3180 (37.7%) | 2730 (48.9%) | 2852 (51.1%) | 2534 (88.5%) | 328 (11.5%) |
| Risk Factors | | | | | | |
| Obesity (n = 79635) | 68205 (85.6%) | 11430 (14.4%) | 16846 (62.7%) | 10030 (37.3%) | 51359 (97.3%) | 1400 (2.7%) |
| Smoking (n = 30818) | 27001 (87.6) | 3817 (12.4%) | 5449 (61.4%) | 3426 (38.6%) | 21552 (98.2%) | 5221 (1.8%) |

men, Indigenous peoples and patients treated in the public sector died. A higher proportion of patients with chronic kidney disease (CKD) and chronic obstructive pulmonary disease (COPD) died inside or outside the hospital. The most common comorbidities in all patients with COVID-19 that resulted in death were high blood pressure (20383) and diabetes (17660). Among hospitalized patients, more than 40% of them with any of the four comorbidities analyzed died.

Table 2 shows admissions to hospitals, ICUs, and deaths among Indigenous peoples and non-Indigenous people. A total of 4178 (1.0%) of patients with COVID-19 were Indigenous. Of these, a total of 1602 (38.3%) were admitted to the hospital compared to only 27.2% of non-Indigenous people. Additionally, 155 (3.7%) Indigenous peoples were admitted to the ICU compared to only 2.1% of non-Indigenous people. Regarding deaths (outside and inside a hospital), 691 (16.5%) of all Indigenous peoples died. Only 11.1% of non-Indigenous people died. The proportion of deaths among Indigenous peoples was higher compared to non-Indigenous people inside the hospital (37.1% vs 36.3%, respectively) and in the ICU (52.9% and 51.6%). Deaths outside the hospital were also higher among Indigenous peoples (3.7% vs 1.7%).

Table 3 shows the distribution of sociodemographic characteristics and the prevalence of comorbidities and risk factors among Indigenous and non-Indigenous non-survivors. A higher proportion of Indigenous non-survivors died in almost all age groups compared to non-Indigenous people. Men are more likely to die in both Indigenous and non-Indigenous people; however, Indigenous men and women were more likely to die from COVID-19 than non-Indigenous people. While a higher proportion of deaths happened in the public sector, Indigenous peoples were more likely to die in both sectors. Finally, a higher proportion of Indigenous peoples with comorbidities and risk factors died compared to non-Indigenous people.

Regarding our multiple logistic regression models (Table 4), we observed that, compared to non-Indigenous people, Indigenous peoples in general and those who received ambulatory care had significantly higher odds of dying (OR 1.13, 95%CI 1.03 to 1.24 in all COVID-19 patients; OR 1.55, 95%CI 1.24 to1.92 in non-hospitalized patients). Interestingly, people treated in the public health care sector also had higher odds of death compared to people treated in the private sector in all the models. However, this was the only factor where the OR was higher among hospitalized patients than non-hospitalized patients (OR 3.22, 95%CI 2.91 to 3.55 vs OR 2.14, 95%CI 1.72 to 2.71). Men, older people, patients with obesity and patients with comorbidities had higher odds of dying as well, especially among non-hospitalized patients.

## Discussion

To our knowledge, this is the most extensive study of hospital and ambulatory mortality of COVID-19 patients in Mexico. Among all patients, we found that patients who died were more likely to be older, men, Indigenous, receive treatment in the public sector, and have

**Table 2. Hospital admission, ICU admission and deaths by ethnicity.**

|  | Indigenous n = 4178 (1.0%) | Not Indigenous n = 412368 (99.0%) | Total n = 416546 |
|---|---|---|---|
| Admitted to hospital | 1602 (38.3%) | 112251 (27.2%) | 113853 |
| Admitted to ICU | 155 (3.7%) | 8715 (2.1%) | 8870 |
| Deaths | 691 (16.5%) | 45817 (11.1%) | 46508 |
| % deaths (hospital) | 37.1 | 36.3 | - |
| % deaths (not in hospital) | 3.7 | 1.7 | - |
| % deaths (ICU) | 52.9 | 51.6 | - |

**Table 3. Sociodemographic characteristics and present comorbidities and risk factors among all non-survivors by ethnicity.**

|  | Indigenous n = 691 | Non-Indigenous n = 45817 |
|---|---|---|
| Age (years ± SD) | 63.4 ± 13.0 | 61.9 ± 13.8 |
| Age group |  |  |
| 18–39 | 28 (2.3%) | 2724 (1.7%) |
| 40–49 | 73 (8.8%) | 5828 (6.2%) |
| 50–59 | 153 (18.0%) | 10705 (14.0%) |
| 60–69 | 94 (13.6%) | 12599 (26.4%) |
| 70 or older | 243 (39.6%) | 13961 (39.2%) |
| Gender |  |  |
| Female | 230 (13.6%) | 16128 (8.3%) |
| Male | 461 (18.5%) | 29689 (13.6%) |
| Sector |  |  |
| Private | 4 (12.1%) | 561 (4.9%) |
| Public | 687 (16.6%) | 45256 (11.3%) |
| Comorbidities |  |  |
| Diabetes | 258 (28.7%) | 17402 (25.9%) |
| COPD | 47 (34.6%) | 2192 (33.7%) |
| HBP | 270 (29.6%) | 20113 (24.0%) |
| CKD | 40 (44.4%) | 3140 (37.6%) |

comorbidities and lifestyle risk factors. Similar to previous studies, we found that age, gender, and the presence of comorbidities are important predictors of death among patients with COVID-19 [1, 2, 4, 5, 9–11]. Overall, Mexicans have a high burden of chronic diseases such as diabetes, high blood pressure, and chronic kidney disease, although the prevalence of some of these diseases seems to be higher among Indigenous peoples [14–18]. Our results show that a higher proportion of Indigenous peoples in almost all age groups, in both genders, and with comorbidities died in our sample, which suggests access to care and the quality of care might play a role on the impact of COVID-19 among Indigenous peoples.

We found that ethnicity was mostly associated with death in those who received ambulatory care even after adjusting for health care sector. This finding suggests that poor access to health care services might be a driver of higher mortality among Indigenous peoples. Despite progress in the socioeconomic and health conditions of Indigenous peoples in recent years, most of them remain uninsured or are affiliated to the public health care sector, which is often of poorer quality compared to the private sector [12, 19–21]. Furthermore, one study found that Indigenous peoples in Mexico were less likely to receive any type of health care among those seeking medical care [22]. The current COVID-19 pandemic has strained health care systems worldwide. Therefore, an inequitable distribution of health resources could be playing a role in access to care as well. For example, municipalities with a high proportion of Indigenous peoples have only 63 clinics, 31 beds and 86 doctors per 100000, while those with a low proportion of Indigenous peoples have 377 clinics, 336 beds and 670 doctors per 100000 [12]. Finally, language barriers may also affect access to care at the local level. This is particularly relevant to older Indigenous peoples, where only 1 out of 5 report speaking Spanish in addition to their Indigenous languages [12].

Our study had some limitations. Selection bias could be present as Mexico is mainly using a sentinel surveillance system to identify and report COVID-19 cases. This system mainly identifies people seeking care. Thus, asymptomatic and mild cases might be missed. This could be

**Table 4. Multiple logistic regression models (ORs and 95%CIs) on all COVID-19 patients, hospitalized patients and non-hospitalized patients.**

|  | All COVID-19 patients | Hospitalized patients | Non-hospitalized patients |
|---|---|---|---|
| Age group |  |  |  |
| 18–39 (ref) | 1 | 1 | 1 |
| 40–49 | 3.38 (3.22–3.54)** | 1.82 (1.72–1.92)** | 3.53 (3.11–4.02)** |
| 50–59 | 7.40 (7.09–7.74)** | 2.79 (2.65–2.94)** | 8.21 (7.29–9.27)** |
| 59–69 | 15.05 (14.40–15.73)** | 4.13 (3.92–4.34)** | 18.62 (16.51–21.05)** |
| 70 or older | 27.20 (26.00–28.47)** | 5.95 (5.64–6.26)** | 38.31 (33.92–43.39)** |
| Gender |  |  |  |
| Women (ref) | 1 | 1 | 1 |
| Men | 1.79 (1.75–1.83)** | 1.36 (1.32–1.39)** | 2.09 (1.97–2.22)** |
| Ethnic group |  |  |  |
| Non-indigenous (ref) | 1 | 1 | 1 |
| Indigenous | 1.13 (1.03–1.24)** | 0.92 (0.83–1.02) | 1.55 (1.24–1.92)** |
| Sector |  |  |  |
| Private (ref) | 1 | 1 | 1 |
| Public | 2.89 (2.64–3.16)** | 3.22 (2.91–3.55)** | 2.14 (1.72–2.71)** |
| Comorbidities* |  |  |  |
| Diabetes | 1.61 (1.57–1.65)** | 1.17 (1.14–1.21)** | 1.72 (1.61–1.84)** |
| COPD | 1.36 (1.29–1.44)** | 1.12 (1.06–1.20)** | 1.72 (1.45–1.98)** |
| High blood pressure | 1.27 (1.24–1.30)** | 1.14 (1.11–1.17)** | 1.30 (1.21–1.38)** |
| Chronic kidney disease | 2.57 (2.44–2.71)** | 1.66 (1.59–1.79)** | 3.02 (2.64–3.44)** |
| Risk Factors* |  |  |  |
| Obesity | 1.39 (1.35–1.42)** | 1.17 (1.13–1.20)** | 1.62 (1.52–1.73)** |
| Smoking | 0.99 (0.95–1.03) | 0.98 (0.94–1.03) | 0.93 (0.84–1.04) |

*The reference groups were those without the comorbidity or risk factor.

**$p < 0.05$.

particularly problematic for our sample of patients who did not require hospitalization and, thus, our whole sample as well. Ethnicity was also missing in 12562 (2.9%) of 432501 adult patients with COVID-19. However, this percentage is much lower than previous studies conducted in the UK and Brazil [4, 5]. Ethnicity was determined in this data set by asking the patient if they spoke an Indigenous language. According to a recent national survey, 21.5% of Mexicans consider themselves Indigenous, but only 6.5% of the population actually spoke an Indigenous language [23]. Only 4178 (1.0%) patients spoke an Indigenous language in our sample, which suggests Indigenous peoples lack access to adequate testing and are, therefore, underrepresented in our sample. Finally, our models were not able to be adjusted for socioeconomic status. For this reason, we included health care sector as a variable in our model unlike previous studies in Mexico, as income is associated with insurance type.

Mexico is a racially and ethnically diverse country. However, most Mexicans are considered *mestizos*, or people of mixed European, African and Indian ancestry. This has resulted in the erasure of racial identities and it poses a challenge to the study of race in the country [24]. However, some studies have attempted to measure health inequities by using skin color rather than race self-identity. For example, one study showed that light brown, dark brown and black Mexicans had lower levels of self-rated health compared to White Mexicans [25]. We were only able to classify ethnicity as Indigenous and non-Indigenous based on language. Further research is need on how the COVID-19 pandemic has affected non-White Mexicans in general, regardless of self-identify.

Our study also has several strengths. To our knowledge, our study is the first study to address the impact of a pandemic in a specific vulnerable group, such as Indigenous peoples in Mexico. Our study adds to the growing evidence of how ethnic and racial inequities affect the health of REM across different settings in the world. Another key strength of the study was the large sample size with relatively few missing data for all the predictors that were analyzed. The large sample size also enabled high statistical power which yielded narrow confidence intervals for our multiple regression model. Our study also accounted for the health care sector in which patients were treated, which had not been included in previous studies conducted in Mexico.

In conclusion, we present evidence suggesting that Indigenous peoples in Mexico have a higher risk of death from COVID-19. Our results also suggest that access to care might be playing an important role on the impact of COVID-19 among Indigenous peoples. The COVID-19 pandemic has highlighted what we have known for quite a long time: Indigenous peoples and ethnic minorities continue to be marginalized, and urgent action is needed to address the health inequities that persist among the most vulnerable. Beyond addressing the current inequities in the COVID-19 response, the financial, social, and educational barriers need be addressed as well if we hope to achieve social and health justice for Mexican Indigenous peoples and other ethnic minorities.

## Supporting information

**S1 Data.**
(CSV)

## Author Contributions

**Conceptualization:** Ismael Ibarra-Nava, Magaly Padilla-Orozco.

**Data curation:** Ismael Ibarra-Nava, Kathia G. Flores-Rodriguez.

**Formal analysis:** Ismael Ibarra-Nava, Kathia G. Flores-Rodriguez.

**Investigation:** Ismael Ibarra-Nava, Violeta Ruiz-Herrera, Hilda C. Ochoa-Bayona, Alfonso Salinas-Zertuche, Magaly Padilla-Orozco, Raul G. Salazar-Montalvo.

**Methodology:** Ismael Ibarra-Nava, Violeta Ruiz-Herrera, Hilda C. Ochoa-Bayona, Alfonso Salinas-Zertuche, Raul G. Salazar-Montalvo.

**Project administration:** Ismael Ibarra-Nava.

**Resources:** Raul G. Salazar-Montalvo.

**Supervision:** Hilda C. Ochoa-Bayona, Magaly Padilla-Orozco, Raul G. Salazar-Montalvo.

**Validation:** Kathia G. Flores-Rodriguez, Hilda C. Ochoa-Bayona.

**Visualization:** Ismael Ibarra-Nava, Kathia G. Flores-Rodriguez.

**Writing – original draft:** Ismael Ibarra-Nava.

**Writing – review & editing:** Ismael Ibarra-Nava, Hilda C. Ochoa-Bayona, Raul G. Salazar-Montalvo.

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
