## [Decision Letter · Decision Letter 0]

11 Nov 2020

PONE-D-20-29414

Ethnic disparities in COVID-19 mortality in Mexico: a cross-sectional study based on national data

PLOS ONE

Dear Dr. Raul Salazar-Montalvo,

Thank you for submitting your manuscript to PLOS ONE. After careful consideration, we feel that it has merit but does not fully meet PLOS ONE’s publication criteria as it currently stands. Therefore, we invite you to submit a revised version of the manuscript that addresses the points raised during the review process.

Please address each comment made by the reviewer.

We look forward to receiving your revised manuscript.

Kind regards,

Mary Hamer Hodges, MBBS MRCP DSc

Academic Editor

PLOS ONE

Additional Editor Comments:

This is a much needed paper on COVID in Mexico. However the analysis does not add a great deal to the debate but could take the analysis and discussion to a more complex level.

Journal Requirements:

2. In ethics statement in the manuscript and in the online submission form, please provide additional information about the patient records used in your retrospective study. Specifically, please ensure that you have discussed whether all data were fully anonymized before you accessed them and/or whether the IRB or ethics committee waived the requirement for informed consent. If patients provided informed written consent to have data from their medical records used in research, please include this information.

Reviewers' comments:

Reviewer's Responses to Questions

**Comments to the Author**

1. Is the manuscript technically sound, and do the data support the conclusions?

Reviewer #1: Yes

2. Has the statistical analysis been performed appropriately and rigorously? 

Reviewer #1: I Don't Know

3. Have the authors made all data underlying the findings in their manuscript fully available?

Reviewer #1: Yes

4. Is the manuscript presented in an intelligible fashion and written in standard English?

Reviewer #1: Yes

5. Review Comments to the Author

Reviewer #1: This is an important and much needed paper and it really is excellent to see ethnic and gender stratification of data on COVID in Mexico. As the authors point out, COVID in Mexico is tracking the same lines as elsewhere with ethnic minority groups experiencing worse outcomes and more men dying than women.

I think however that the analysis does not add a great deal to the debate and our understandings of COVID as it stands, it could however take the discussion and analysis to a more complex level.

1) Greater clarity is needed about the use of terms race, ethnicity and indigenous. Indigenous is a category defined culturally and not biologically. It is not a category of race, but of social defined characteristics such as history and language. As the authors state, in the case of these data indigeneity is defined as a speaker of indigenous language. In this case it would be good to a) state up front what the national defininition is -are the national statistics on indigeneity also based solely on lanaguage? If not, can you state what the defintion is (as opposed to the 'language' indicator in the COVID) data.

2) In relation to the above, can you clarify the relationship between ethnicity and COVID mortality. For example, is language a barrier to care, are indigenous people less likely to seek care, are there different transmission patterns in indigenous localities and what are these.

3) What about Afro-Mexican populations. Are there any data on the incidence and morbidity of COVID among other non-indigenous ethnicities?

4) Consider if you actually need to use the term 'race' and whether this just muddies the water. There are no solid data (that I am aware of) that suggest a biological risk factor for speficic racial characteristics, so race is really a proxy for social inequality. If this is the case, it should be stated and explained clearly.

5) What data are there on testing? So, for example, are indigenous people overrepresented in the mortality and hospitalisations data because they are testing less? Some context on this would help. So for example with such a high rate of uninsured indigenous people and the excessive cost of testing in the private sector, the higher death rate could be due to the lower testing rate of indigenous people. Some mention of consideration of this is important to avoid this potential confounder.

6) In relation to 5), what is known about reporting of death rates and COVID in indigenous communities? Can you say anything about the localities where indigenous patients have died? Is there likely to be underrreporting?

6) Could you include some three way tables, for example, to look at the proportion of women, men and indigenous people who receive private care?

6. PLOS authors have the option to publish the peer review history of their article (what does this mean?). If published, this will include your full peer review and any attached files.

Reviewer #1: **Yes: **Jennie Gamlin

---

## [Author Response · Author response to Decision Letter 0]

11 Jan 2021

Reviewer 1 – Jennie Gamlin

1. Greater clarity is needed about the use of terms race, ethnicity and indigenous. Indigenous is a category defined culturally and not biologically. It is not a category of race, but of social defined characteristics such as history and language. As the authors state, in the case of these data, indigeneity is defined as a speaker of indigenous language. In this case it would be good to a) state up front what the national definition is -are the national statistics on indigeneity also based solely on language? If not, can you state what the definition is (as opposed to the 'language' indicator in the COVID data)?

Response: Dear Ms. Gamlin. First of all, thank you for your comments and questions. They have been of great help to us. Regarding this first comment, we agree that being Indigenous is defined culturally and not biologically. For this reason, we decided to add the three definitions in our manuscript (please see page 6-7, lines 127-133). We also added a new reference to support this (see reference Our analyses only included those who spoke an Indigenous language because that was the only variable available in the data set. Fortunately, this is the definition most commonly used in Mexico. We agree that being Indigenous is not a category of race, although Indigenous people suffer from a lot of racism in Mexico. We were more careful throughout the article with the use of these two words. 

2. In relation to the above, can you clarify the relationship between ethnicity and COVID mortality. For example, is language a barrier to care, are indigenous people less likely to seek care, are there different transmission patterns in indigenous localities and what are these.

Response: Unfortunately, we were unable to determine if there are different transmission patterns in Indigenous localities because we had no access to this type of data. Furthermore, as far as we know, there are not available sources that report statistics specific to Indigenous communities and localities. Regarding access to care, one article did mention language barriers as a potential barrier. We added this to our discussion as we believe it is important to mention (see page 14, lines 240-243). We also discuss that only 1% of the total sample spoke an Indigenous language (compared to 6.5% of the population who report speaking an Indigenous language). This suggests an inequitable distribution of resources, testing and access to care (see page 14, lines 254-257). Thank you. 

3. What about Afro-Mexican populations. Are there any data on the incidence and morbidity of COVID among other non-indigenous ethnicities?

Response: Thank you for this question. We believe it is an extremely important one. Unfortunately, we do not “measure” race the same way other countries do. We discuss why in page 15, lines 260-269. To this date, no studies have been done that attempt to compare skin color and COVID-19 outcomes. 

4. Consider if you actually need to use the term 'race' and whether this just muddies the water. There are no solid data (that I am aware of) that suggest a biological risk factor for specific racial characteristics, so race is really a proxy for social inequality. If this is the case, it should be stated and explained clearly.

Response: Thank you for this suggestion. We chose to include the term race in our discussion because the literature uses both “race” and “ethnicity” as potential determinants of health in the COVID-19 pandemic. We understand that it is not being used in the literature to imply biological differences, but rather as a way to highlight social inequities, as you mention. In our article, we were careful with the use of the term “race”. In fact, we only mention it as a “self-identity” in the Mexican context in the discussion (see page 15, lines 262-266). 

5. What data are there on testing? So, for example, are indigenous people overrepresented in the mortality and hospitalisations data because they are testing less? Some context on this would help. So, for example with such a high rate of uninsured indigenous people and the excessive cost of testing in the private sector, the higher death rate could be due to the lower testing rate of indigenous people. Some mention of consideration of this is important to avoid this potential confounder.

Response: Thank you for this question. The data set we used included all the people who had been tested up until August 3rd, 2020. As mentioned in lines 254-257, Only 4178 of our sample (COVID-19 positive) spoke an Indigenous language. We checked the total sample to include those who tested negative and only 9037 spoke an Indigenous language (around 1% of all those who were tested). This is considerably lower than the 6.5% of the Mexican population who speak an Indigenous language. We mention this is a limitation in page 15, lines 254-256 as well. 

6. In relation to 5), what is known about reporting of death rates and COVID in indigenous communities? Can you say anything about the localities where indigenous patients have died? Is there likely to be underrreporting?

Response: Thank you. This is a great question. Unfortunately, no data on this is available in the literature or in official government sources. The data set we used includes “municipality of residence” as a variable. However, this does not mean that the person actually received attention there. The Mexican health system is using a sentinel surveillance system to diagnose COVID-19 cases, which means not every facility is actually testing suspicious patients. As mentioned above, we do believe there is underreporting because of the small proportion of Indigenous peoples being tested and diagnosed with COVID-19 (both around 1%).

---

## [Decision Letter · Decision Letter 1]

27 Jan 2021

PONE-D-20-29414R1

Ethnic disparities in COVID-19 mortality in Mexico: a cross-sectional study based on national data

PLOS ONE

Dear Dr. %Raul G. Salazar-Montalvo%,

Thank you for submitting your manuscript to PLOS ONE. After careful consideration, we feel that it has merit but does not fully meet PLOS ONE’s publication criteria as it currently stands. Therefore, we invite you to submit a revised version of the manuscript that addresses the points raised during the review process.

please address the study limitations identifed by the reviewer

We look forward to receiving your revised manuscript.

Kind regards,

Mary Hamer Hodges, MBBS MRCP DSc

Academic Editor

PLOS ONE

Additional Editor Comments (if provided):

Please add in a 'limitation' section which picks up on the unanswerable questions (see original review comments as well) and considers the data in this light and comment on whether any new data has emerged since this paper was written which have any impact on the findings, since the situation is rapidly changing.

Reviewers' comments:

Reviewer's Responses to Questions

**Comments to the Author**

1. If the authors have adequately addressed your comments raised in a previous round of review and you feel that this manuscript is now acceptable for publication, you may indicate that here to bypass the “Comments to the Author” section, enter your conflict of interest statement in the “Confidential to Editor” section, and submit your "Accept" recommendation.

Reviewer #1: (No Response)

2. Is the manuscript technically sound, and do the data support the conclusions?

Reviewer #1: Yes

3. Has the statistical analysis been performed appropriately and rigorously? 

Reviewer #1: Yes

4. Have the authors made all data underlying the findings in their manuscript fully available?

Reviewer #1: Yes

5. Is the manuscript presented in an intelligible fashion and written in standard English?

Reviewer #1: Yes

6. Review Comments to the Author

Reviewer #1: The authors have adequately responded to the concerns of the first review although, as they note in their response, some of the concerns cannot be addressed due to lack of data. For example the extent of underreporting is not know, the low level of testing may skew results, it is not clear where in the country data correspond to. There is further concerns that the indigenous respondents only make up 1% of the dataset, which means that they are very considerably underrrepresented in data. This being the case it would be helpful if the authors could add in a 'limitation' section which picks up on these unanswerable questions (see original review comments as well) and considers the data in this light. I also wonder if there are any new data which have emerged since this paper was written which have any impact on the findings, since the situation is rapidly changing it would be worth thinking about this.

7. PLOS authors have the option to publish the peer review history of their article (what does this mean?). If published, this will include your full peer review and any attached files.

Reviewer #1: No

---

## [Author Response · Author response to Decision Letter 1]

6 Feb 2021

Thank you both for your feedback and suggestions. We have made further changes to our limitations section in the Discussion. (See page 14, lines 248-252 and page 15, lines 263 and 267). Just an update regarding our testing strategy: it has not changed, unfortunately. The sentinel surveillance system is still in place, and the government has not announced any new plan regarding testing or focus on Indigenous communities. Furthermore, we thought it would be important to mention the lack of trust Indigenous communities have on the government, as this week an entire community of 40,000 Indigenous peoples are refusing to vaccinate because they do not trust the government.

---

## [Editor Report · Decision Letter 2]

15 Feb 2021

Ethnic disparities in COVID-19 mortality in Mexico: a cross-sectional study based on national data

PONE-D-20-29414R2

Dear Dr. % Raul Salazar-Montalvo%,

We’re pleased to inform you that your manuscript has been judged scientifically suitable for publication and will be formally accepted for publication once it meets all outstanding technical requirements.

Kind regards,

Mary Hamer Hodges, MBBS MRCP DSc

Academic Editor

PLOS ONE

Additional Editor Comments (optional):

Thank you for these revisions.
---

## [Editor Report · Acceptance letter]

23 Feb 2021

PONE-D-20-29414R2 

Ethnic disparities in COVID-19 mortality in Mexico: a cross-sectional study based on national data 

Dear Dr. Salazar-Montalvo:

I'm pleased to inform you that your manuscript has been deemed suitable for publication in PLOS ONE. Congratulations! Your manuscript is now with our production department. 

Kind regards, 

on behalf of

Dr. Mary Hamer Hodges 

Academic Editor

PLOS ONE